# Interleukin-33-Enhanced CXCR4 Signaling Circuit Mediated by Carcinoma-Associated Fibroblasts Promotes Invasiveness of Head and Neck Cancer

**DOI:** 10.3390/cancers13143442

**Published:** 2021-07-09

**Authors:** Yu-Chun Lin, Wen-Yen Huang, Tsai-Yu Lee, Yi-Ming Chang, Su-Feng Chen, Yaoh-Shiang Lin, Shin Nieh

**Affiliations:** 1Department of Pathology, National Defense Medical Center & Tri-Service General Hospital, Taipei 11490, Taiwan; yuchunlin@mail.ndmctsgh.edu.tw (Y.-C.L.); yiminnn@mail.ndmctsgh.edu.tw (Y.-M.C.); 2Graduate Institute of Medical Sciences, National Defense Medical Center, Taipei 11490, Taiwan; 3Department of Radiation Oncology, Tri-Service General Hospital, National Defense Medical Center, Taipei 11490, Taiwan; hwyyi@mail.ndmctsgh.edu.tw; 4Division of Colon and Rectum Surgery, Department of Surgery, Tri-Service General Hospital Songshan Branch, National Defense Medical Center, Taipei 10581, Taiwan; TSGH88165@mail.ndmctsgh.edu.tw; 5School of Dentistry, China Medical University, Taichung 404333, Taiwan; 6Department of Otorhinolaryngology, Head and Neck Surgery, Kaohsiung Veterans General Hospital, Kaohsiung 813414, Taiwan

**Keywords:** head and neck squamous cell carcinoma, carcinoma-associated fibroblast, interleukin-33/CXCR4 regulatory circuit, tumor microenvironment, tumor progression

## Abstract

**Simple Summary:**

The tumor microenvironment (TME) plays an important role of cancer recurrence and treatment resistance. The cytokines in the TME may involve the tumor pathogenesis. Therefore, it is critical to discover the novel TME-associated signal transduction pathways contributing for the tumor progression. In this study, we established stable clones of interleukin (IL)-33-overexpressing head and neck squamous cell carcinoma (HNSCC) cells to simulate IL-33-induced autocrine signaling and identify the role of IL-33 expression in more aggressive phenotypes with increased mobility and properties of cancer stemness to provide a potential therapeutic strategy for HNSCC patients, on the basis of targeting the IL-33-enhanced-CXCR4 regulatory circuit.

**Abstract:**

Despite recent advances, treatment for head and neck squamous cell carcinoma (HNSCC) has limited efficacy in preventing tumor progression. We confirmed previously that carcinoma-associated fibroblasts (CAF)-induced interleukin-33 (IL-33) contributed to cancer progression. However, the molecular mechanisms underlying the complex communication network of the tumor microenvironment merited further evaluation. To simulate the IL-33-induced autocrine signaling, stable clones of IL-33-overexpressing HNSCC cells were established. Besides well-established IL-33/ST2 and SDF1/CXCR4 (stromal-derived factor 1/C-X-C motif chemokine receptor 4) signaling, the CAF-induced IL-33 upregulated CXCR4 via cancer cell induction of IL-33 self-production. The IL-33-enhanced-CXCR4 regulatory circuit involves SDF1/CXCR4 signaling activation and modulates tumor behavior. An in vivo study confirmed the functional role of IL-33/CXCR4 in tumor initiation and metastasis. The CXCR4 and/or IL-33 blockade reduced HNSCC cell aggressiveness, with attenuated invasions and metastases. Immunohistochemistry confirmed that IL-33 and CXCR4 expression correlated significantly with disease-free survival and IL-33-CXCR4 co-expression predicted a poor outcome. Besides paracrine signaling, the CAF-induced IL-33 reciprocally enhanced the autocrine cancer-cell self-production of IL-33 and the corresponding CXCR4 upregulation, leading to the activation of SDF1/CXCR4 signaling subsequent to cancer progression. Thus, targeting the IL-33-enhanced-CXCR4 regulatory circuit attenuates tumor aggressiveness and provides a potential therapeutic option for improving the prognosis in HNSCC patients.

## 1. Introduction

Head and neck squamous cell carcinoma (HNSCC) was the seventh most common cancer in 2018 accounting for over 890,000 new cases and 450,000 deaths worldwide [1]. Despite early detection and advances and improvements in treatment, there has been an unsatisfactory 5-year survival rate of approximately 50% [2]. While most early-stage HNSCCs can be cured with surgical resections, advanced HNSCCs are difficult to eliminate using surgery alone, and multiple combined treatments that include chemotherapy and radiotherapy combined with surgery are required. A subpopulation of cancer cells called cancer stem-like or stem cells (CSCs) has been shown to exhibit therapeutic resistance leading to local recurrences and distant metastases. CSCs are a promising target for cancer treatment, based on their therapeutic resistance; however, identifying CSCs within the heterogenic malignant tumor can be difficult, and there is currently no simple biomarker for their identification [3]. These drawbacks necessitate novel treatment strategies to target the tumor stroma and the tumor itself. The tumor stroma offers an attractive target for cancer therapy and may provide a new path for therapeutic intervention.

It has been reported that HNSCCs and other types of tumors frequently activate or enhance mediators such as cytokines or chemokines that are generated from stroma components to promote tumor growth and progression. Chemokines are a family of chemoattractant cytokines that play an important role in the control of cell migration in several different human cancers [3,4]. They also promote the tumor cell movement required for metastasis [5]. The receptor CXCR4, and its ligand CXCL12 (also called SDF-1), is the key chemokine essential for HNSCC metastasis. A study has shown that there was significant activation of the CXCR4 in HNSCCs with regard to cell growth, differentiation, and survival and that the CXCR4 was significantly higher in patients with positive lymph node involvements and distant metastases [6]. The impact of chemokines and their receptors on HNSCCs may be described by their influence on matrix metalloproteinases (MMPs) [7]. Moreover, tumor cells can stimulate stromal cells to synthesize and secrete growth-promoting chemokines, establishing a reciprocal cross tumor–stromal interaction that favors tumor growth. Accomplice cells in the tumor microenvironment (TME), such as cancer-associated fibroblasts (CAFs) and macrophages have also been shown to produce chemokines and promote tumor growth [8,9,10].

A cytokine, interleukin 33 (IL33), released by CAFs, has been reported to induce treatment failure in HNSCCs [11] and promote the stemness properties of via crosstalk between tumor cells and CAFs [12]. In our previous study, we used an organotypic culture to investigate CAFs that promote the aggressive behavior of cancer cells and the paracrine effect of CAF-induced IL-33, which increases the aggressiveness of HNSCCs [13]. However, the pathogenesis of this complex tumor–stromal signaling remains to be completely elucidated. A better understanding of the mechanisms of the complex communication network of the TME via IL-33 in tumor–stromal signaling, on a cellular and molecular basis, may provide a new potential therapeutic target for the treatment of HNSCCs.

Therefore, we aimed to establish stable clones of the IL-33-overexpressing HNSCC cells and an animal model to confirm whether IL-33 exerts an autocrine effect on cancer cells in addition to the paracrine effect of CAF-induced IL-33, which increases the aggressiveness of HNSCC.

## 2. Results

### 2.1. Establishment of Stable Clones of the IL-33-Overexpressing HNSCC Cells and Characterization According to Their Aggressive Phenotypes 

To simulate the autocrine effect of IL-33, we first established two stable clones, FaDu and TW204, which are HNSCC cells that overexpress IL-33. PCR and Western blotting analysis demonstrated that stable clones of the IL-33-overexpressing HNSCC cells with ample functional IL-33 were related to the enhanced expression of the corresponding ST2 (Figure 1a,b). Through flow cytometry, a higher level of CXCR4 expression was found in stable clones of the IL-33-overexpressing HNSCC cells (Figure 1c).

### 2.2. Generation of CAFs with a Different Feature from HGFs and the Provision of a Functional Role of the Paracrine Effect in HNSCC Cells

To address the functional roles of the CAFs in TME, we compared two representative CAFs and human gingival fibroblasts (HGFs). On examining the phenotypic differences between the CAFs and HGFs, we found that α-SMA positive CAFs were relatively larger in size and secreted more SDF1 and IL-33 than HGFs (Figure 1d). Furthermore, the flow cytometry analysis showed that the primarily cultured CAFs displayed a higher expression of CD10 (>80%) and GPR77 (>65%) than the HGFs (3.3% and 1.9%, respectively) (Figure 1e). Using a 3D-organotypic raft culture, we found that the CAFs induced invasion of the HNSCC cells was more aggressive than that of the HGFs via a paracrine pathway. However, no obvious invasion induction was seen with the HGFs, which lacked the paracrine signaling (Figure 1f). The invasion ability was tested among the five groups using 3D organotypic raft cultures. Invasion was indicated by the presence of infiltrative foci in the HNSCC cells. No obvious invasion inductions were seen with the HGFs, which lacked the paracrine signaling characteristic of IL-33 in the control HNSCC cells. The depth of the tumor invasions showed a statistically significant increase in the groups with both the IL-33-overexpressing stable clones in sections of the matrix layer with embedded CAFs compared to the conditioned groups of the IL-33-overexpressing stable clones following blockages of IL-33 or CXCR4, which showed less invasion depth in the HNSCC cells.

### 2.3. Stable Clones of IL-33-Overexpressing HNSCC Cells with Autocrine Activities Related to Cancer Progression Because of the “IL-33/CXCR4 Regulatory Circuit”

Western blotting analysis demonstrated that stable clones of the IL-33-overexpressing HNSCC cells with ample functional IL-33 were related to enhanced CXCR4 expression. Inhibition of IL-33 decreased CXCR4 expression; however, blockade at the level of CXCR4 did not influence IL-33 expression (Figure 2a, left). A similar result was evident with the inhibition of IL-33, which resulted in the decreased expression of invasion-related MMP2 and MMP9 (Figure 2a, right). Increased migration and invasion abilities were observed in the cloned IL-33-overexpressing HNSCC cells compared with the abilities of the control and the HNSCC cells with IL-33 and CXCR4 blockades (Figure 2b,c). Western blotting analysis showed that blockade of either IL-33 or CXCR4 decreased the number of molecules involved in the p38MAPK signaling cascade (Figure 2d). However, the blockade of p38MAPK, ERK1/2, and JNK did not affect the expression of IL-33 but CXCR4 expression was affected by p38MAPK blockade (Figure 2e).

### 2.4. Autocrine and Paracrine Effects of IL-33 Contribution to the Induction of Cancer Stem Cell (CSC) Properties, Radioresistance, Chemoresistance, and Antiapoptosis

To understand the molecular level of the crosstalk between the cancers and CAFs involving IL-33, we confirmed the autocrine and paracrine effects of IL-33 on the expression of CSC properties (Figure 3a–c), including chemoresistance, radioresistance, alterations in drug resistance genes, and antiapoptosis (Figure 3d,e). Stably cloned cells were used to evaluate the autocrine effect of the HNSCC cells that arose from the IL-33 autosecretion. Cells exposed to the recombinant IL-33 were used to study the paracrine effect of the IL-33 on HNSCC cells. The IL-33-overexpressing HNSCC cells and HNSCC cells exposed to the recombinant IL-33 showed an increased expression of CSC-representative markers, sphere formations, and significant increases in both radiosensitivity and chemoresistance, compared with the control HNSCC cells. Furthermore, both the autocrine and paracrine effects of IL-33 were involved in the upregulation of the drug resistance-related genes, *ABCG-2* and *MDR-1*, and the increased expression of antiapoptotic proteins.

### 2.5. IL-33 Enhancement of the Capabilities of Tumorigenicity and Metastasis in Stably Cloned FaDu Cells In Vivo 

To confirm the tumor-initiating capabilities, both the stable clones of the IL-33-overexpressing FaDu cells and the control cells were injected subcutaneously into BALB/c nude mice, and the tumorigenicity of the transplants was analyzed. A greater capability for tumorigenicity with significant differences in gradual growth in a time-dependent manner was observed in the stable clone of the IL-33-overexpressing FaDu cells compared with the control cells (Figure 4a,b). Both the stable clones of the IL-33-overexpressing FaDu cells and the control cells were injected into the tail vein of BALB/c nude mice to test their metastatic potential. The stable clone of the IL-33-overexpressing FaDu cells exhibited a greater capability for metastasis than did the control cells. Immunohistochemistry showed that both IL-33 and CXCR4 were reciprocally positive in both newly formed subcutaneous tumors and metastatic lung nodules (Figure 4c,d).

### 2.6. Correlation of the IL-33 Expression with the CXCR4 Expression and the Relationship to the Outcome

To examine the clinical significance of the relationship between the expression of IL-33 and CXCR4, we analyzed samples from 40 representative HNSCC patients with available clinical information, as described previously [13]. Consistent with the results of IL-33 in cancer cells, CXCR4 expression varied regardless of the invasion pattern grading score (Figure 5a). The linear regression analysis showed a close relationship between IL-33 and CXCR4 expression (Figure 5b). The immunohistochemistry of the tissue samples confirmed that there was a significant correlation between IL-33 and CXCR4 expression and disease-free survival, and that the co-expression of IL-33 and CXCR4 predicted a poor outcome in the Kaplan–Meier analysis (Figure 5c). However, IL-33 and CXCR4 co-expression did not show a significant correlation with clinical parameters, including TNM stage, except for tumor differentiation (Figure 5d).

## 3. Discussion

The stromal cells in the surrounding TME, such as the CAFs, not only act as an active contributor to cancer initiation but also contribute actively to cancer progression [14,15,16]. A recent report indicated that targeting IL-33/ST2 signaling was a therapeutic strategy for inflammatory disorders. Another study claimed that IL-33 was used as a target for patients with treatment-resistant breast cancer via the attenuation of epithelial–mesenchymal transition and cancer stemness [17,18]. However, as described earlier, immune cell-associated IL-33 might conversely attenuate tumor progression and provide an option for immunotherapy [19]. A review of the literature showed a paucity of reports involving IL-33/ST2 or SDF1/CXCR4 signaling to target TME for tumor progression. Instead of functioning initially as an important inflammatory cytokine, the CAF-induced IL-33 expression was responsible for enhancing cancer cell progression via a paracrine effect [12]. In this study, we found relevant evidence in a step-by-step manner to elucidate the molecular mechanisms underlying the complex communication network between the CAFs and the cancer itself, between the cancer cells, and their transduction signaling axis, which can lead to cancer cell progression with the induction of CSC properties. 

We first proposed to determine the nature and origins of the CAFs [20]. The CAFs in the TME were generated and transformed from the HGFs. There was a progressive transition that showed functional and morphological alterations between the HGFs and CAFs. Due to the diversity of transformed fibroblasts, the primarily cultured CAFs displayed a higher expression of CD10 and GPR77 than the HGFs, which expressed lower levels of CD10 and GPR77, according to the findings of a previous study [21]. These findings indicated that the increased CD10 and GPR77 expression implicated a transitional tendency when the HGFs turned into CAFs. The higher expressions of CD10 and GPR77 corresponded to the increased likelihood of HGFs becoming CAFs. Moreover, the CAFs possessed a greater ability to functionally induce HNSCC cells to be more aggressive than the HGFs, as shown by the 3D organotypic raft culture. The induction of invasion was markedly increased by the CAF-induced IL-33 expression via the paracrine effect. 

Identifying invasion-related molecules and possible mechanisms that block their pathways can play a crucial role in the diagnosis and treatment of HNSCCs. A recent study [22] proposed that the IL33-TGF beta niche signaling promoted cancer progression. Furthermore, two recent reports have suggested that the immune cell-associated and proinflammatory cytokine IL-33 might conversely attenuate tumor progression [23,24]. However, another study reported the opposite result, and the controversy about the functional role of IL-33 in tumor–stromal signaling, still exists [19]. By establishing stable clones of IL-33-overexpressing HNSCC cells, we confirmed the presence of high IL-33 levels in the HNSCC cells. A higher expression of ST2 and CXCR4 was also found in the cloned IL-33-overexpressing HNSCC cells. These data suggested that the established stable clones of IL-33-overexpressing HNSCC cells acted in an autocrine manner and displayed an aggressive phenotype that was closely associated with a higher ST2 and CXCR4. Both the IL-33/ST2 and SDF1/CXCR4 signaling axes are well-established molecular modules. Nevertheless, the results of this study revealed a special interaction of collateral signaling between IL-33 and CXCR4, besides the classical IL-33 linkage to ST2. This novel finding was validated by driving role that the functional CAF-induced IL-33 plays in cancer cells, to regulate CXCR4 expression, wherein the autocrine cancer-secreting IL-33 automatically target the surface receptor ST2 and, subsequently, enhance the IL-33/ST2 signaling cascade transcriptional upregulation of CXCR4, which then generates the newly discovered “IL-33/CXCR4 regulatory circuit,” which undergoes repeated amplification for further activation of the SDF-1/CXCR4 signaling transduction. A higher expression of CXCR4 receptors was found in the cloned IL-33-overexpressing HNSCC cells, along with increased levels of MMP2 and MMP9. A blockade of either IL-33 or CXCR4 decreased the invasion ability. However, the cloned IL-33-overexpressing HNSCC cells together with the autocrine cancer-secreting function increased MMP2 and MMP9 expression spontaneously and, conversely, the blockade of the IL-33 expression enhanced the attenuation of MMP2 and MMP9, suggesting that IL-33 was an upstream mediator of the signaling cascade. The shutdown of IL-33 and CXCR4 elicited the opposite results. Moreover, the inhibition of IL-33 decreased the number of molecules involved in the p38MAPK and pNFkb signaling cascade. The inhibition of CXCR4 did not have a similar effect, again suggesting that IL-33, rather than CXCR4, was an upstream mediator of the signaling cascade. However, using the inhibitors of the phosphorylated p38MAPK, ERK1/2, and JNK did not affect the expression of IL-33 but decreased CXCR4 expression. In this study, we further examined the expression of CSC properties, including the expression of other CSC representative markers, sphere-forming abilities, radioresistance and chemoresistance, drug-resistant genes (*ABCG2* and *MDR-1*)*,* and antiapoptotic proteins in the IL-33-overexpressing stable clones and HNSCC cells exposed to recombinant IL-33. Thus, all the most important properties of stemness induced by either the paracrine or autocrine IL-33 signaling were enhanced. The IL-33/CXCR4 regulatory circuit was reconfirmed by in vivo experiments. These study data demonstrated a high capability of tumorigenicity and metastases secondary to the induction of the IL-33-overexpressing stably cloned FaDu cells, leading to CXCR4 upregulation in vivo. Immunohistochemistry showed that both IL-33 and CXCR4 were reciprocally positive in both the newly formed subcutaneous tumors and metastatic lung nodules.

A high CXCR4 expression in cancer cells may be an independent predictor of lymph node metastases and poor survival in colon cancer patients [25,26]. A literature review showed that under special conditions, such as marked inflammation, necrosis, or hypoxia, or even in cancer progression. Interestingly, a recent report has shown that HIF-1α enhances IL-33 production by enabling certain signaling pathways, especially the p38MAPK and ERK pathways. IL-33, by turn, induces HIF-1α expression, thus forming a HIF-1α/IL-33 positive feedback loop [27]. Furthermore, TNF-α promotes HIF-1α expression by both triggering NF-κB signaling and controlling IL-33 expression in a HIF-1α-dependent manner. Therefore, TNF-α facilitates HIF-1α-dependent IL-33 expression on the basis of identifying a functional binding site for HIF-1α in the IL-33 promoter region [28]. Although the mechanisms are still not clear, these discoveries suggest that IL-33 may be upregulated in hypoxic tumor microenvironments. Both IL-33 and hypoxia-inducible factor 1 (HIF-1)-related CXCR4 levels were markedly increased. This HIF-dependent activation of CXCR4 that is mediated by IL-33 may suggest a novel mechanism for the induction of tumor progression. To confirm the clinical significance of the relationship between the expression of IL-33 and CXCR4, we analyzed samples from 40 representative HNSCC patients whose clinical information was available. Consistent with the results of the clinical tissue samples, the linear regression analysis showed a close relationship between IL-33 and CXCR4 expression. The immunoexpression of IL-33 and CXCR4 showed a significant correlation with disease-free survival, and the co-expression of IL-33 and CXCR4 represented a worse outcome. However, the co-expression of IL-33 and CXCR4 showed a significant correlation with clinical parameters, including TNM stage, nodal involvement, stage, and tumor differentiation.

It is noteworthy that many hypotheses have been proposed to explain the nature and origin of the CAFs [29,30,31]. In this study, we observed that the dual role of the CAFs is mediated by IL-33, that is, normal supporting fibroblasts that are called persevered or putative CAFs for the maintenance of hemostasis, once involved in carcinogenesis, will through their crosstalk with cancer cells, trigger the IL-33/CXCR4 signaling circuit and can contribute to tumor aggressiveness. Finally, the complex crosstalk with plentiful molecular communications proceeds like a symphony that is orchestrated and conducted by both autocrine and paracrine signaling between CAFs and tumor. The diagrammatic illustrations demonstrate our proposal to explain the major mechanism and the relationships between the molecules involved in the IL-33-p38MAPK-CXCR4-SDF-1 loop between the TME and the actual tumor (Appendix A). Extending these data to clinical practice suggests that modulating the TME by targeting the IL-33/CXCR4 signaling circuit may attenuate the aggressiveness of cancer cells and provide a novel therapeutic strategy in precision medicine to improve the prognosis in HNSCC patients.

## 4. Materials and Methods

### 4.1. Cell Cultures and Reagents

Stromal fibroblasts separated from adjacent carcinoma cells and uninvolved intraoral gingival tissues were obtained from five patients undergoing surgical resections of HNSCCs. The CAFs and HGFs were isolated using differential trypsinization with a modified protocol [12]. This was followed by histological confirmation. The cell line was tested for mycoplasma contamination (Mycoplasma Reagent Set; Euroclone s.p.a, Pavia, Italy). Both fibroblast types were maintained in Dulbecco’s modified Eagle’s medium (DMEM) and Ham’s nutrient mixture F12 culture medium, supplemented with 10% fetal bovine serum (FBS) and 1% penicillin–streptomycin. The FaDu (ATCC HTB-43) and TW204 (from a Chinese nasopharyngeal carcinoma in Taiwan) cell lines [32], originated from the human hypopharynx geal and nasopharyngeal squamous cell carcinomas, were cultured in a RPMI-1640 medium supplemented with 10% FBS, 2  mM glutamine, 1% penicillin/streptomycin, 1% sodium pyruvate, and 1% of amino-acid solution. Cells were treated for 6  h with recombinant IL33 protein with doses of 50 and 25 μg/mL AMD3100 (Sigma, Zwijndrecht, The Netherlands) [33] and then, in a series of experiments, were pre-treated for 1  h with various enzymatic inhibitors of different signaling pathways: 50  μM FR180204 (ERK, inhibitor), 10 μM SB203580 (p38MAPK inhibitor) or 4 μM SP600125 (JNK inhibitor).

### 4.2. Viral Production and Infection of Target Cells

The entire coding sequence was excised from the pMSCV vector (Appendix A) and subcloned into the mammalian expression vector, pMSCV-IL33. The final construct was verified by sequencing. The positive clones were selected using ampicillin and amplified in bacterial culture media. All the plasmids were purified using the Endo-free Plasmid Mini Kit II (OMEGA). For transfection, FaDu and NPC204 cells were plated in six-well plates (Corning, Lowell, MA, USA), cultured in a complete growth medium to 80% confluence, and then cultured in a medium without fetal bovine serum (FBS) for 12–16 h. Transfection was performed using 1 μg (per dish) of either p pMSCV or p pMSCV–IL33 and Turbofect (Thermo Fisher Scientific Inc. Branchburg, NJ, USA) reagent, according to the manufacturer’s protocol.

### 4.3. Western Blot, RNA Extraction and Quantitative Real-Time PCR

The antibodies and primers used are listed in Appendix A. Western blotting was performed according to a standard protocol, as described previously [12]. For the RNA extraction, the total RNA from the cultured cells was extracted using TRIzol (Invitrogen Life Technologies, Carlsbad, CA, USA) and 1 µg RNA was used for the cDNA synthesis. Quantitative real-time PCR was performed to quantify the gene expression, using the StepOnePlus real-time PCR system (Applied Biosystems, San Francisco, CA, USA). Data from three independent experiments were performed in triplicate and were expressed as mean ± SD.

### 4.4. Migration and Invasion Assay

The in vitro transwell migration and invasion assays were performed and lower wells were coated with 15 μg/mL collagen type I, incubated for 1 h at 37 °C, and blocked overnight with phosphate-buffered saline (PBS) containing 1% bovine serum albumin at 4 °C. After the blocking buffer was removed, the lower wells were loaded with 300 μL of 10^−7^ M CXCL12 in serum-free RPMI. The cells were serum-starved overnight and harvested with enzyme-free cell detaching buffer. The inserts were loaded with 2 × 10^4^ cells in 150 μL per condition and were allowed to migrate for 4.5 h at 37 °C. After migration, the nonmigrated cells were removed with a cotton swab wetted in PBS. Cells at the bottom surface were fixed in 75% methanol for 20 min at room temperature, stained with 0.25% Coomassie blue in 45% for 20 min at room temperature, washed, air-dried, and mounted on a microscope slide. The number of migrated cells was calculated by counting the cells from 10 fields of view per slide, using a counting grid and 40× magnification. Data from three independent experiments were performed in triplicate were expressed as mean ± SD.

### 4.5. Organotypic 3D Culture

For Organotypic 3D culture, eight volumes of collagen I/Matrigel (collagen I:Matrigel = 1:1) were mixed with 1 volume of 10× DMEM and 1 volume of FBS containing fibroblasts (5 × 10^5^). The gel mixture was dispensed into a 12 mm Millicell insert (Millipore, Bedford, MA, USA) inserted into a six-well culture plate. The mixture was allowed to set at 37 °C for 24 h, and the FaDu cells (2 × 10^5^) were seeded atop the gel mixture. After a 24 h incubation period, the cancer cells were exposed to air by removing the medium from the surface. The gel was then fed from underneath with the complete medium, which was changed daily. After 21 days, the cultured tissue was fixed and embedded in paraffin for a histological examination. Data from three independent experiments were performed in triplicate were expressed as mean ± SD.

### 4.6. Flow Cytometry

The 1 × 10^6^ single-cell suspension from trypsinized cells and spheres were added in 1 mL phosphate-buffered saline (PBS) and stained with aldehyde dehydrogenase 1 (ALDH1; ALDEFLUOR assay kit; Stem Cell Technologies, Durham, NC, USA). After labeling, the cells were washed with PBS three times and stained subsequently with FITC- or PE-labeled secondary antibody for 30 min, in the dark. After three cycles of washing with PBS, FaDu and TW204 cells were incubated with 1:100 polyclonal rabbit anti-hCXCR4 antibody (Abcam) or with PBS (2.7 mM KCl, potassium 1.8 mM KH_2_PO_4_, 137 mM NaCl, 10.1 mM Na_2_HPO_4_, pH 7.4) for 45 min on ice, followed by 30 min of incubation with mouse–anti rabbit antibody phycoerythrin-labeled (Southern Biotech, Uithoorn, The Netherlands). The cells were analyzed using a flow cytometer, FACSCalibur (Epics Elite; Coulter Electronics, Mijdrecht, The Netherlands). Data analysis was performed using Kaluza software (Beckman Coulter Nederland BV, Woerden, The Netherlands).

### 4.7. Enzyme-Linked Immunosorbent Assay and Sphere Culture

The medium from the confluent normal fibroblast and the CAF were sampled at 48 h after plating and centrifuged to remove the cell debris. SDF-1 and IL-33 levels in medium were assayed using the Quantikine Human IL-33 and SDF1 Immunoassay kit (R&D Systems, Abingdon, United Kingdom) according to the manufacturer’s instructions. The measured levels of IL-33 were expressed as picograms and SDF-1 as per 1 mg of protein in the cell lysate. The supernatants were collected and analyzed using ELISA, according to the manufacturer’s standard protocol. ELISAs were performed in duplicate on three separate occasions and the data were expressed as means ± SDs.

### 4.8. Assays for Chemosensitivity and Radiosensitivity 

The cells were seeded in a 10 cm dish at a density of 1×10^6^ cells/dish. For the chemosensitivity assay, the cells were treated with 0–30 μM cisplatin (Sigma, St. Louis, MO, USA) for 48 h. For the radioresistance assay, cells were irradiated using a CyberKnife radiosurgery system (Accuray, Sunnyvale, CA, USA) that could deliver different doses (2–10 Gy). The relative survival fraction of the cells was determined using an MTS assay with the Cell Titer 96 Aqueous One Solution Cell Proliferation Assay kit.

### 4.9. In Vivo Assay

All the animal experiments were approved by the Institutional Animal Care and Use Committee of the National Defense Medical Center (IACUC-13-039). The in vivo tumorigenicity study was performed according to the guidelines of the local ethics committee that had full accreditation awarded to it by the Association for Assessment and Accreditation of Laboratory Animal Care in the National Defense Medical Center. There were five mice in each group. The mice were kept at 18–26 °C, in 30–70% humidity, and independently airconditioned under a 12 h dark/12 h light cycle for seven days before the xenograft injection. The parental FaDu-Ctrl and FaDu-IL33 cells were injected into BALB/c nude mice (five weeks). The cell suspension (100 μL) was injected subcutaneously into each mouse with different tumor cell numbers, from 1 × 10^6^, 1 × 10^5^ and 1 × 10^4^ cells. The tumors formed seven days after injection. The tumor sizes were monitored and measured weekly according to the formula (length × width2)/2. At 30 days after the orthotopic inoculation, the mice were euthanized under anesthesia using chloroform. To evaluate the metastatic capability, stable clones and control cells were injected intravenously into the tail vein. The cancer cells were harvested at a concentration of 1 × 10^6^ cells per 0.1 mL DPBS. A volume of 0.1 mL of the suspended cells was injected through the tail vein into 5-week-old male BALB/c mice using 27-gauge needles. The mice were sacrificed and examined for the growth of metastatic tumors at various time points. The mice injected with FaDu-Ctrl or FaDu-IL33 cells were sacrificed seven weeks after injection. Two independent experiments were performed for each pair of cancer cells. At seven weeks, the mice were euthanized under anesthesia.

### 4.10. Human Tumor Tissue Collection and Immunohistochemical Staining 

Archival tissue specimens from the primary tumors and lymph nodes were obtained from the Tri-Service General Hospital in the Taiwan between 2009 and 2013. Immunohistochemistry was performed on paraffin-embedded sections of the HNSCC specimens. After deparaffinization and dehydration, specimens were brought to the boil in 10 mM sodium citrate buffer (pH 6.0) for 40 min for antigen retrieval and then blocked in peroxidase-blocking solution (Dako Cytomation, Glostrup, Denmark). Rabbit monoclonal anti-CD133 antibody (clone C24B9, Cell Signaling Technology, Danvers, MA, USA) was diluted to 1:200 and incubated at 4 °C overnight. The staining was detected using an Envision detection system (peroxidase/DAB+, rabbit/mouse, Dako Cytomation). The specimens were counterstained with Mayer’s hematoxylin. CXCR4 expression was assessed by staining with IL33 (R & D) and CXCR4 antibody (Abcam), secondary goat anti-rabbit antibody conjugated to peroxidase (DAKO, Heverlee, Belgium), and the subsequent tertiary rabbit anti-goat was conjugated to peroxidase (DAKO). Staining was visualized by 3,3′-diaminobenzidine.

Only clinical cases without any neoadjuvant androgen deprivation were selected. All the tissue specimens were encoded with unique numbers. According to Dutch law, no further institutional review board approval was required (www.federa.org (accessed on 4 July 2016)). The FFPE tissue specimens were mounted on slides as whole tissue sections and stained with hematoxylin and eosin. We chose one of the most representative and adequate sections from the total positive tissue samples and then counted for intensity and distribution. For immunohistochemistry, the treated but without the primary antibody specimen was used as a negative control, and a human retina with immunoreactivity was used as a positive control. The intensities of IL33 and CXCR4 immunoreactivity of the tumor cells were classified into four categories: 0 (no staining), 1 (weak staining), 2 (moderate staining), and 3 (strongest intensity). The distribution was also measured as the percentages of the positively stained tumor cells (from 0 to 100%) in the total tumor volumes in each section. To compare immunoexpression for each case, the percentage of positive cells of intensity was multiplied by the corresponding intensity to obtain an immunoreactivity score ranging from 0 to 300. The staining results were evaluated independently by two pathologists blinded to the patients’ clinical information. Discrepancies between the pathologists were resolved by consensus.

### 4.11. Statistical Analysis

The independent Student’s *t*-test or ANOVA was used to compare the continuous variables between the groups and the Chi-square test was used for the comparison of the dichotomous variables. The level of statistical significance was set at a *p*-value < 0.05. All the statistical analyses were performed using SPSS version 20 (SPSS, Inc., Chicago, IL, USA).

## 5. Conclusions

These data provide an insight into the mechanisms underlying the interactions of CAFs with cancer cells via both the autocrine and paracrine signaling of IL-33. Targeting the interleukin-33/CXCR4 signaling circuit attenuated cancer aggressiveness and may have potential as a treatment strategy for improving the prognosis of HNSCC patients. Further research should be performed to ascertain the exact role of the IL-33/CXCR4 axis and to obtain evidence-based knowledge for HNSCC treatment.

## Figures and Tables

**Figure 1 cancers-13-03442-f001:**
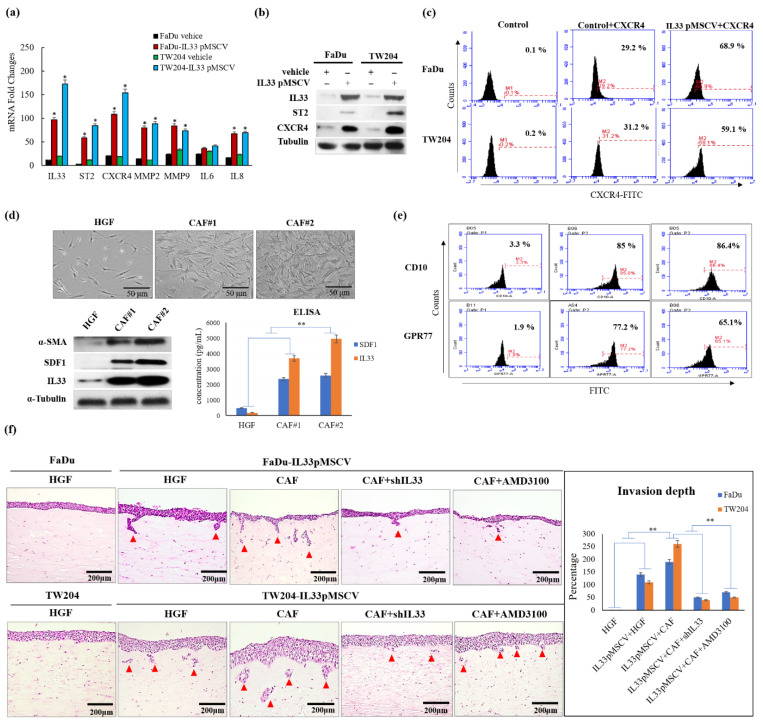
Comparative analysis and characterization of CAFs and HGFs and their effects on two stable clones of the IL-33-overexpressing HNSCC cells and control HNSCC cells. (**a**) Besides the upregulation of IL-33, ST2, and CXCR4, the expressions of two representative invasion biomarkers, MMP2 and MMP9 by RT-PCR, were enhanced statically, in the cloned IL-33-overexpressing HNSCC cells compared to in the control HNSCC cells. (**b**) Western blotting analysis showed higher expressions of ST2 and CXCR4 in the cloned IL-33-overexpressing HNSCC cells than in the control HNSCC cells. The uncropped Western blots have been shown in Appendix A. (**c**) Flow cytometry revealed higher expressions of CXCR4 in the cloned IL-33-overexpressing HNSCC cells than in the control HNSCC cells. (**d**) The top panel shows the phenotypic differences between the CAFs and HGFs. The Western blotting analysis showed that the expressions of SDF-1 and IL33 in two representative CAFs were statistically higher than that in HGFs (*p* < 0.05) with further validation by ELISA shown at the bottom panels. (**e**) Flow cytometry demonstrated that the CAFs displayed higher expressions of CD10 and GPR77 than the HGFs. (**f**) Four test groups and one control group were classified for a comparative analysis of invasive ability via organotypic culture. The induction of invasion (shown by arrows) demonstrated that there were many foci of the infiltrated cells in the IL-33-overexpressing stable clones, which were markedly enhanced by the CAF-induced IL-33 (in upper and lower center panels). However, in the control group and one test group wherein the IL-33-overexpressing stable clones were cultured without the CAF-induced IL-33, no obvious invasion induction or decreased invasion induction was seen (left two upper and lower panels). The blockade of the IL-33 via shIL-33 or by using an inhibitor of CXCR4 (25 µg/mL AMD3100) markedly decreased the invasion ability (right upper and lower two panels). The statistical analysis of tumor invasion depth is shown in the right panel. * *p* < 0.05; ** *p* < 0.01.

**Figure 2 cancers-13-03442-f002:**
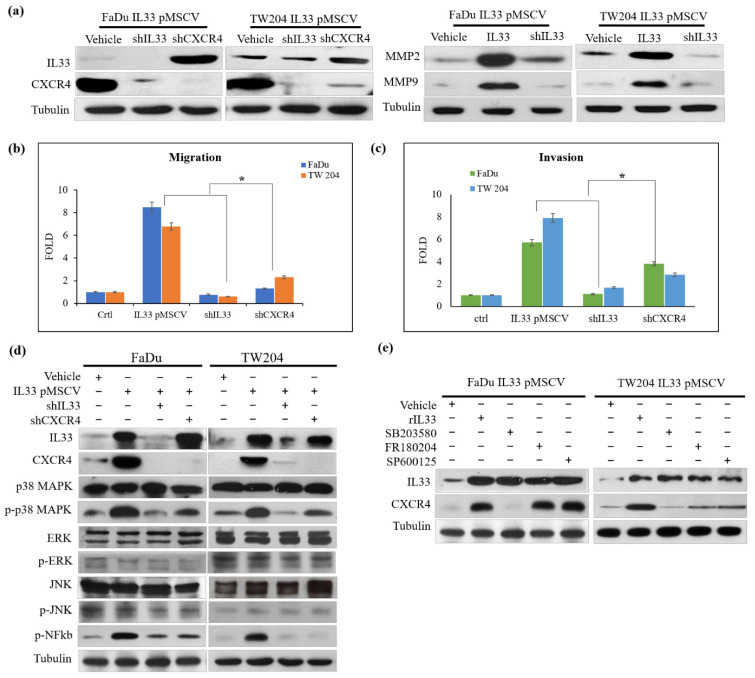
Comparative analysis of invasion-related biomarkers associated with the expression of IL-33/CXCR4 signaling between two stable clones of the IL-33-overexpressing HNSCC cells and control. (**a**) Western blotting analysis demonstrated that increased expressions of IL-33 and CXCR4 were seen in both the IL-33-overexpressing stable clones. Inhibition by shIL-33 markedly decreased the expression of CXCR4; however, the shutdown of CXCR4 did not decrease the expression of IL-33, suggesting that IL-33 expression was mediated upstream of CXCR4 in the signaling cascade. Increased expressions of invasion-related MMP2 and MMP9 were seen in the IL-33-overexpressing stable clones; conversely, shutdown of IL-33 markedly decreased the expression of MMP2 and MMP9. Increased migration (**b**) and invasion (**c**) abilities were seen in the cloned IL-33-overexpressing HNSCC cells compared with those of the controls. Blockades with IL-33 and CXCR4 elicited the opposite results. (**d**) Western blotting analysis showed that blockade of the IL-33 expression decreased the molecules involved in the p38MAPK and pNFκB signaling cascade. In contrast, the inhibition of CXCR4 did not induce similar effects, again suggesting that IL-33 was situated upstream of CXCR4 in the signaling cascade. (**e**) However, the inhibition of phosphorylated p38 through ERK1/2 and JNK did not affect IL-33 expression, but it decreased CXCR4 expression. * *p* < 0.05.

**Figure 3 cancers-13-03442-f003:**
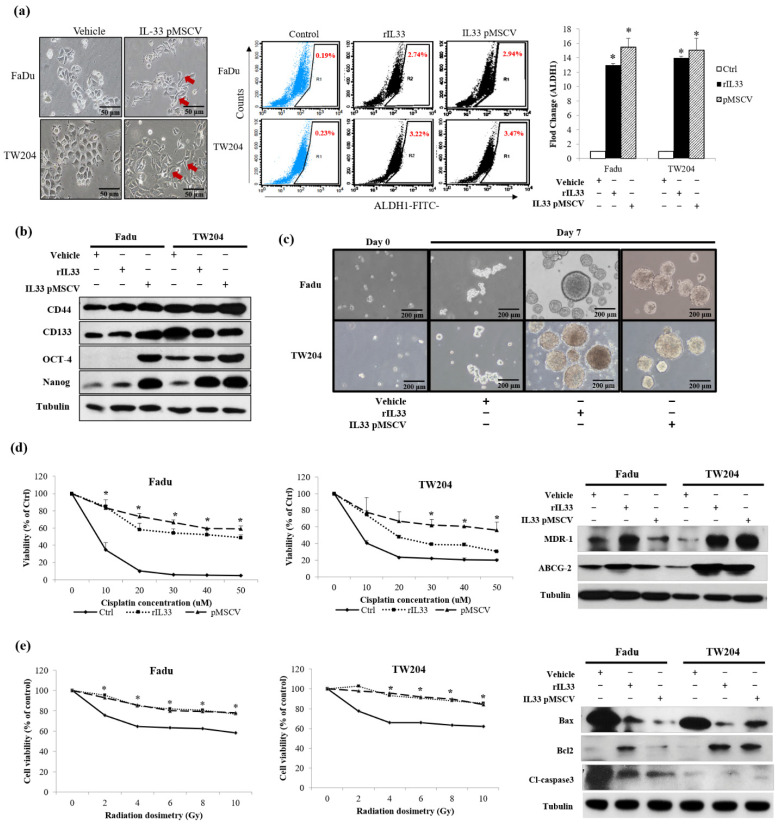
Characterization of the two stably cloned IL-33-overexpressing HNSCC cells facilitated by an additional IL-33 expression that led to the induction of the CSC properties. ((**a**), left) A more aggressive phenotype was seen in the stable clones of the IL-33-overexpressing HNSCC cells (arrows) acting in an autocrine manner, compared with the parent cells. Using the recombinant IL-33 for the paracrine effect of the CAF-induced IL-33, showed a similar effect. There were significant differences in both the stably cloned HNSCC cells and HNSCC cells exposed to the recombinant IL-33 compared with the control HNSCC cells (*p* < 0.05). ((**a**), middle and right) Flow cytometric analysis of ALDH1 showed significantly increased activity in both the stably cloned HNSCC cells and HNSCC cells exposed to the recombinant IL-33 compared with the control HNSCC cells (*p* < 0.05). (**b**) Western blotting analysis showed the overexpression of CSC-representative markers, including CD44, CD133, Oct-4, and Nanog, in both the stably cloned HNSCC cells and HNSCC cells exposed to the recombinant IL-33 compared with the control HNSCC cells. (**c**) In the nonadherent sphere culture system, both the stably cloned HNSCC cells and the HNSCC cells exposed to the recombinant IL-33 gradually generated well-formed spheres within seven days—an important property of CSC. However, the control cells yielded only discrete, small cell masses without sphere formation. (**d**) Chemoresistance was prominent in both the stably cloned HNSCC cells and the HNSCC cells exposed to recombinant IL-33 compared with the control HNSCC cells. Chemosensitivity did not differ significantly between both the stably cloned HNSCC cell lines and the HNSCC cells exposed to recombinant IL-33 compared with the control HNSCC cells (*p* < 0.05). Western blotting analysis demonstrated that both the stably cloned HNSCC cell lines and the HNSCC cells exposed to the recombinant IL-33 showed an upregulation of the drug resistance-related genes, *ABCG-2* and *MDR-1*. (**e**) Radioresistance was more prominent in both the stably cloned HNSCC cells and the HNSCC cells exposed to the recombinant IL-33 than in the control HNSCC cells. Radiosensitivity differed significantly between both the stably cloned HNSCC cell lines and the HNSCC cells exposed to the recombinant IL-33 compared with the control HNSCC cells (*p* < 0.05). Western blotting analysis demonstrated that both the stably cloned HNSCC cell lines and the HNSCC cells exposed to the recombinant IL-33 showed increased antiapoptotic activity, as shown by the upregulation of Bcl-2 and the downregulation of Bax and caspase 3 compared with the HNSCC cells. * Significant difference (*p <* 0.05) between the experimental and control groups; results are expressed as mean ± SD.

**Figure 4 cancers-13-03442-f004:**
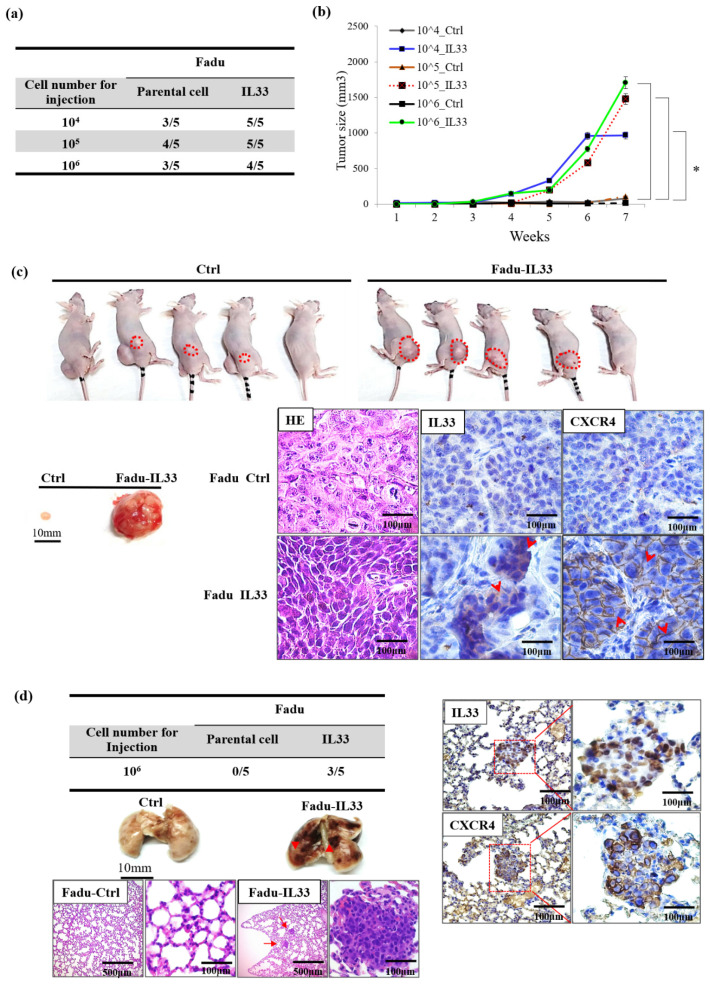
Xerographic evidence from the stable clones of the IL-33-overexpressing HNSCC cells highlights that tumorigenicity and metastases is associated with IL-33 expression/CXCR4 signaling. (**a**) The in vivo study (*n* = 5 in each group) showed greater tumorigenicity in the IL-33-overexpressing stably cloned FaDu cells compared with the control cells. (**b**) Time-dependent tumor growth differences were observed in the stably cloned FaDu cells compared with the control cells. (**c**) The stably cloned cell-induced tumors (right) were significantly larger than the tumors induced by the control cells (left). A marked size difference in the tumors dissected from a representative BALB/c nude mouse was found following the subcutaneous inoculation of the stably cloned FaDu cells and the control cells. The immunohistochemical analysis of the tumor nodules revealed the overexpression of both IL-33 and CXCR4 in the stably cloned FaDu cells (magnification, ×200). (**d**) The in vivo study (*n* = 5 in each group) showed a greater capability for metastasis (3/5) in the IL-33-overexpressing stably cloned FaDu cells than in the control cells (0/5). The gross appearance of the lung tissue dissected from a representative BALB/c nude mouse showed foci of hemorrhagic nodules (arrowheads) in the stable clone cell-induced tumors (right) and no metastatic nodules in the control lung tissue (left). The microscopic findings of the small metastatic hemorrhagic nodules (arrows) in the lung tissue revealed aggregates of many abnormal HNSCC cells with a high nucleus/cytoplasm ratio and a hyperchromatism of the nuclei in the lung parenchyma. Further immunohistochemical analysis of a metastatic tumor nodule in the lung tissue which confirmed their metastatic origin from the HNSCC cells and the overexpression of IL-33 and CXCR4 in the tumor nodules (magnification, ×100 and ×200). In addition, the metastatic small nodules were found in both lungs, and there was no metastases-related involvement of any other organ on necropsy. * *p* < 0.05.

**Figure 5 cancers-13-03442-f005:**
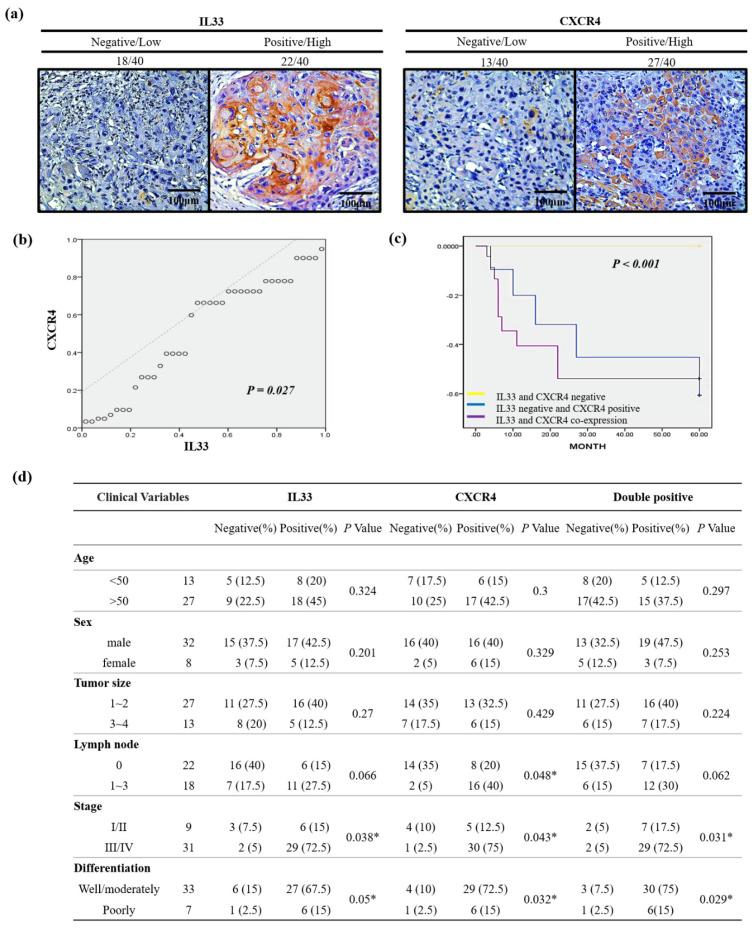
Immunohistochemical analysis of IL-33 and CXCR4 expression and the clinical implications of the IL-33/CXCR4 signaling circuit in 40 patients with HNSCC. (**a**) Photomicrographs of immunohistochemical staining IL-33 and CXCR4 expression in the HNSCC patients. Twenty-two patients (55%) showed negative or low expression of IL-33, 18 patients (45%) showed a positive immunoreactivity of IL-33 expression, 13 patients (32.5%) showed negative or low expression of CXCR4, and 27 patients (67.5%) showed positive immunoreactivity of CXCR4 (magnification, ×200). (**b**) The linear regression analysis of IL-33 and CXCR4 expression showed a significant correlation of IL-33 expression with CXCR4 expression (*p* = 0.027). (**c**) The Kaplan–Meier analysis of disease-free survival of HNSCC patients after a 60-month follow-up plotted as patients who showed negative immunoreactivity versus positive immunoreactivity for IL-33 and CXCR4 expression in tumor cells. Patients negative for IL-33 and CXCR4 expression had a longer disease-free survival compared with those who tested negative for IL-33 and positive for CXCR4. The co-expression of IL-33 and CXCR4 in HNSCC cells showed a significant correlation with shorter disease-free survival (*p* < 0.001) which indicated a worse outcome. (**d**) Moreover, the individual expression of IL-33 and CXCR4 as well as the co-expression of both IL-33 and CXCR4 showed a significant correlation with the clinical parameters including tumor differentiation, lymph node involvement, and TNM stage. * *p* < 0.05.

## Data Availability

The data in the study may be available on request to the corresponding author.

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
