# Peer review of "Interleukin-33-Enhanced CXCR4 Signaling Circuit Mediated by Carcinoma-Associated Fibroblasts Promotes Invasiveness of Head and Neck Cancer"

_cancers, 2021, doi:10.3390/cancers13143442_

Round 1

Reviewer 1 Report

The manuscript ” Interleukin-33-Enhanced CXCR4 Signaling Circuit Mediated by Carcinoma-Associated Fibroblasts Promotes Invasiveness of Head and Neck Cancer” by Yu-Chun Lin et al. investigates the underlying mechanisms of interactions of cancer associated fibroblast with cancer cells via both the autocrine and paracrine signaling of IL-33. The authors conclude that targeting the interleukin-33 / CXCR4 signaling circuit attenuated cancer aggressiveness and may have potential as a treatment strategy to improve the prognosis of HNSCC patients.

In general:
- The paper is technically sound
- The claims are convincing
- The claims are on the whole supported by the experimental data presented
- The claims are appropriately discussed in the context of previous literature
- The manuscript is clearly written and the English is sufficed

However, some issues need to be addressed:
- Please provide the number of experimental and technical replicates for all experiments.
-Specify the statistical methods that was used for the different assays. Did the authors perform any post hoc test for ANOVA analysis? Which?
-Before using abbreviations in the text, the term needs to be introduced (for example MMPs, TME etc)

-Page 2 line 57: “the tumor and the tumor itself”. DO the authors mean the “tumor stroma”?
- Information about error bars in several places missing. For example, in Figure 1 a, d, f. Did the authors present standard deviation or Standard error of mean?

-What is the reason that several sentences and paragraphs are depicted in red?

-Please provide a reference size bar for all microscopy images.

- Flow cytometry graphs are generally very small. It is extremely difficult to read the axis and the percentage of cell populations (for example figure 1 c, e, figure 3 a)

- Please provide Kaplan Meier survival analysis including median and maximum survival for all groups of the xenograft study.
- The Kaplan Meier analysis for HNSCC patients revels a correlation between IL-33 and CXCR4 expression and poor survival. What is the HVP status of these patients? Is there any correlation with the infection and the expression levels and outcome of disease?

Author Response

We appreciate the reviewer's enthusiastic encouragement and response to the comment are listed in the attached file.

Reviewer 2 Report

The authors have now addressed all my initial comments. They have greatly improved the overall quality of the manuscript with a reorganization of figure panels and a better description of the methods.

Author Response

We appreciate the reviewer's enthusiastic encouragement.

This manuscript is a resubmission of an earlier submission. The following is a list of the peer review reports and author responses from that submission.

Round 1

Reviewer 1 Report

Very well written conducted and presented paper with very interesting results, I think that further investigation and larger series will be required in order to confirm your results 

Reviewer 2 Report

Cancers

The manuscript “Interleukin-33–Enhanced CXCR4 Signaling Circuit Mediated by Carcinoma-Associated Fibroblasts Promotes Invasiveness of Head and Neck Cancer” by Yu-Chun Lin at al. investigates whether the expression of IL33 and CXCR4 leads to a more aggressive tumor type. The authors use IL-33-overexpressing HNSCC cell clones to simulate interleukin (IL)-33-induced autocrine signaling to test in vitro cell mobility, resistance to cisplatin and radiotherapy as well as properties of cancer stemness. In an in vivo mouse model, the authors investigate tumorigenicity and metastatic capability.

Generally:

- The study subject is interesting and novel for the chosen cancer type

- The paper is technically sound

- The majority of the claims are convincing

- The claims are on the whole supported by the experimental data presented

However, several aspects need to be addressed:

Materials and methods
Please provide references that refer to the original article that describes the materials and methods used. At several places the provided reference refers a paper that refers a paper and so on.

Please specify the gender/sex of mice used in the experiments.

Please specify the group size and number animals included in the study. How many animals were included in the in vivo tumorigenicity study and how many groups (of 5 animals) were used in the metastatic capability study?

Please describe which statistical analysis was used for which data set. When performing ANOVA analysis, did the authors perform any post hoc testing?

The number of experimental replicates and sample size needs to be provided for all experiments.

What type of radiation was used and at which dose rate?

Add missing axis label for all flow cytometry results.

Figure 3f, add figure legend. Significance symbols are outside of the displayed graph.

Figure 4a, please specify what was measured and what the numbers (3/5, 4/5 etc.) indicate.

Figure 4b, please provide standard deviation of the measurement displayed in the graph.

Figure 4c, did the authors perform a double tumor model? If so, this should be stated in the material and methods section.

Figure 5c, add label on y-axis and add the curves for both IL33 and CXCR4 mono-expression.

Figure 5d, what was the anatomical side of the disease of the specimens included in the study? Was there any difference in HPV status?

Line 178 ELLISA should be ELISA

Reviewer 3 Report

I have now carefully read and examined the manuscript entitled « Interleukin-33–Enhanced CXCR4 Signaling Circuit Mediated by Carcinoma-Associated Fibroblasts Promotes Invasiveness of Head and Neck Cancer » by Lin et al. In this study, the authors attempt to address the exact contribution of interleukin 33 (IL33) and associated signaling pathways to disease progression in HNSCC. While of potential interest in the field, this study suffers from major conceptual and methodological flaws that preclude to draw robust conclusion from the current set of data. The manuscript must be thoroughly edited to present data in a more logical and clearer way, not only in the different figure panels but also in the text. The manuscript is indeed very difficult to read and follow.

The figure 1 is unclear. The authors show data for fiborblast characterization together with data for the characterization of IL33-overexpressing HNSCC clones. This figure must be reorganized with figure 2. Indeed, it unclear for the reviewer what is the difference between figures 1b and 2a. In figure 1a, the graph for SDF1 concentration levels (ELISA ?) is not described. What is the readout for figure 1d ? The authors indicate in the legend « rate of proliferation » but there are no details in the mat and methods section. In the legend for Figure 1b, the authors state « Small foci of infiltrated cells were observed in sections of the matrix layer with embedded CAFs than in those with embedded HGFs in both stably cloned HNSCC cell lines grown in the organotypic raft culture. However, there was no infiltrate in any control cells. ». Where are the data with control cells ?

line 192: « CAFs with morphological alterations appeared more active than HGFs ». What do the authors mean with « more active » ?

Figure 2 : What is the effect of CAF on the invasion capacities of control FaDu cells ? Any difference between HGF and CAF conditions on IL33-overexpressing cells ? A quantification (and statistical analysis) is needed to draw conclusion about the contribution of either autocrine or paracrine IL33-mediated effects.

Instead of referring to a previous publication (ref 13) many times, the authors must provide enough details here to understand the methods used in the current study. Protocols for flow cytometry analyses (CD10/GPR77 immunostaining proliferation assay) are for instance missing. Idem for shRNA transfection of treatment with pharmacological inhibitors (what is the dose and timing for treatment with AMD3100 ?).

Similar problems of presentation and interpretation are present in figures 3 and 4, thereby highlighting the need for a thorough editing of the figure panels and associated text.

Minor points :

Text editing :

-line 148 : 12h dark/12h light

line 174 : to assess the differences

-line 178 : ELISA

-line 199 : statistically enhanced

Reviewer 4 Report

This article reported that IL-33 and CXCR4 pathways were related to CAF and invasiveness of HN cancer.

Major point

  1. The authors should re-rewrite the methods and results clearly and concisely. The results and figures should be organized appropriately to support the hypothesis.
  2. The evidence for autocrine of IL-33 in cancer cells was not strong. The IL-33 from vector-infected cells does not mean that IL-33 was produced by cancer cells itself in tumor tissue.
  3. Add the methods for the IHC of patients’ samples, survival analysis, and IRB number.

Minor points

  1. Describe more detain for Cells, western blotting, and 3D-organotypic culture.
  2. line 182, The word ‘active’ was not appropriate.
  3. Fig 2 (a) Add the statistical result for the invasiveness.
  4. Fig 2 e. Add the data for the cells with IL-33MSCV which were treated with shIL33 and shCXCR4.
  5. Fig 2 e. Add the blotting data for the effect of inhibitors on the target proteins.